# Solving the Problem of Elasticity for a Layer with N Cylindrical Embedded Supports

Vitaly Miroshnikov [1,*] , Oleksandr Savin [1], Vladimir Sobol [1] and Vyacheslav Nikichanov [2]

1 Department of Aircraft Strength, National Aerospace University "KHAI", 61000 Kharkiv, Ukraine
2 Department of Aircraft Production Technology, National Aerospace University "KHAI", 61000 Kharkiv, Ukraine
* Correspondence: v.miroshnikov@khai.edu; Tel.: +380-67-7893333

**Abstract:** The main goal of deformable solid mechanics is to determine the stress–strain state of parts, structural elements, and their connections. The most accurate results of calculations of this state allow us to optimize design objects. However, not all models can be solved using exact methods. One such model is the problem of a layer with cylindrical embedded supports that are parallel to each other and the layer boundaries. In this work, the supports are represented by cylindrical cavities with zero displacements set on them. The layer is considered in Cartesian coordinates, and the cavities are in cylindrical coordinates. To solve the problem, the Lamé equation is used, where the basic solutions between different coordinate systems are linked using the generalized Fourier method. By satisfying the boundary conditions and linking different coordinate systems, a system of infinite linear algebraic equations is created. For numerical realization, the method of reduction is used to find the unknowns. The numerical analysis has shown that the boundary conditions are fulfilled with high accuracy, and the physical pattern of the stress distribution and the comparison with results of similar studies indicate the accuracy of the obtained results. The proposed method for calculating the stress–strain state can be applied to the calculation of structures whose model is a layer with cylindrical embedded supports. The numerical results of the work make it possible to predetermine the geometric parameters of the model to be designed.

**Keywords:** Lamé equation; embedded supports; cylindrical cavity in a layer; generalized Fourier method

## 1. Introduction

The main goal of designing structures, machine parts, and mechanisms is to optimize dimensions and materials. For this purpose, it is very important to choose the most accurate methods for determining the stress–strain state of a body. However, most 3D models (especially those with more than three boundary surfaces) do not allow for analytical methods. In these cases, such a scheme is simplified, or approximate methods are used, such as the finite element method [1]. However, to use the finite element method, the model must have finite dimensions, which is not always the case. In addition, approximate methods do not give confidence in the final result. Therefore, to improve accuracy, tests of finished elements are performed [2] or a combination of different calculation methods [3]. The use of tests is time-consuming and requires new tests whenever any element parameters change. Therefore, for complex models, it is relevant to find the most effective application, combination, or generalization of existing analytical and analytical-numerical methods.

A layer with cylindrical embedded supports can be represented as a layer with cylindrical cavities on which displacements are given or as a layer with cylindrical rigid inclusions (provided that the loads are balanced). There are many works in the literature on the study of cylindrical stress concentrators in a body. For example, in [4], a plate with a cylindrical hole is considered, where metaheuristic optimization algorithms are used to determine the stress–strain state of a body around stress concentrators. In [5], a semi-analytical polynomial method is used for a similar plate. In the problem, a system of nonlinear algebraic

equations is created that takes into account the nonlinearity in thickness, and then it is solved using the Newton–Raphson method. The bending of rectangular plates with a stress concentrator in the form of a circular cutout is studied in [6] using the complex potential method. However, the methods used in [4–6] can only be applied to a plane problem with a stress concentrator located perpendicular to the body surface.

The spatial problem for a slab with a stress concentrator located perpendicular to the body surface is studied in [7]. The authors investigate the wave field for a layer with a perpendicular cylindrical hole. The solution is based on a combination of the Laplace integral transform and the finite integral sin- and cos-Fourier transforms. However, this method does not allow solving problems for a layer with cylindrical cavities that are parallel to the layer surfaces.

A combination of analytical and experimental methods has been proposed in [8–11]. For example, in [8], a layered composite is considered under impact loading. In the analytical part, the parameters of the displacement vector in each layer are decomposed into a power series along the transverse coordinate. By way of experiment, the authors study the maximum displacement of samples under impact loading using an indenter. In [9], a simple analytical problem is formulated for a similar model in the form of a trigonometric series expansion. To perform this, the original non-canonical shell is embedded in an auxiliary canonical form. An auxiliary form for this is the experimental model [10], which reproduces the process of a bird hitting the surface of a rigid body. An analytical model of multilayer glazing was developed in [11]. This model is a generalization of the first-order theory, which allows for transverse shift deformations. In the experimental part, a mathematical model of an impulsive load is investigated. In the experimental part, the pressure pulse was studied and a mathematical model was developed. The above methods [8–11] can be applied to laminated composites with transverse cylindrical inhomogeneities, but they do not allow to consider longitudinal inhomogeneities.

The problem of determining the stress state in a half-space with longitudinal inhomogeneity in the form of a thick-walled pipe was considered in [12]. In this problem, the generalized Michell solution in a plane system of polar coordinates for radially homogeneous bodies is applied. However, the proposed method does not allow for consideration of the problem in a spatial formulation and cannot sufficiently take into account the layer boundaries. Another approach to solving problems with longitudinal inhomogeneities is the Fourier series expansion and the image method. For example, for a slab with a cylindrical hole, the stress state is calculated based on the Fourier series in [13]. As a result, the stress distribution is obtained with high accuracy. In [14], the method of images is used in the problem of wave diffraction for a slab with a cylindrical hole. However, the methods used in [13,14] can only solve problems for a plane model and take into account a maximum of three boundary surfaces.

The most effective analytic-numerical method for a spatial problem with many boundary surfaces is the generalized Fourier method [15]. The justification of this method in terms of the formulas for the transition of the basic solutions of the Lamé equation between a cylinder and a half-space is presented in [16]. The results of these studies make it possible to transfer the basic solutions of the Lamé equation from one coordinate system to another. However, for the case of transition between the Cartesian and cylindrical systems, only the case when these systems are equally oriented and connected to each other is taken into account.

Thus, in [17], the problem for an infinite cylinder with four cylindrical cavities parallel to each other was solved using the generalized Fourier method. The problem for an elastic cylinder with N parallel cylindrical cavities was solved in [18]. The problem for a model where cylindrical cavities form a hexagonal structure in a cylinder was solved in [19], and for a model where 16 cylindrical inclusions are in an elastic cylinder in [20]. Works [17–20] use formulas for the transition of basic solutions only between different cylindrical coordinate systems. This approach does not allow solving problems for a layer where the Cartesian system is used.

This relationship is applied in [21–27], where the generalized Fourier method is also used. For example, the problem for an elastic half-space with a longitudinal cylindrical cavity was solved in an actually flat formulation in [21] and in a spatial formulation in [16]. However, problems for half-space do not take into account the lower surface of the layer. The lower bound has been considered in the problems for a layer with a single circular cylindrical cavity or a single elastic cylindrical inclusion in [22–26]. Thus, the third basic problem of elasticity theory is considered in [22]. In this problem, the stresses in the cylindrical cavity are set to zero, the lower surface of the layer is rigidly fixed, and the upper surface of the layer is subjected to a non-zero load. Paper [23] solves the second basic problem of elasticity theory with a given periodic load. Paper [24] solved the problem of an infinite elastic layer with an elastic solid cylindrical inclusion and given displacements on the layer surface. In [25], stresses are set on the layer surfaces, and zero displacements are set on the inner surface of an embedded cylindrical pipe. In [26], a calculation methodology is proposed, and a numerical analysis of the stressed state is performed for a bilayer structure with a cylindrical void between parallel flat surfaces. However, papers [22–26] do not take into account the formulas for the transition between basic solutions from one cylindrical coordinate system to another, which does not allow solving problems with more than one cylindrical stress concentrator. The formulas for two cylindrical stress concentrators are considered in [27], where a method for solving the problem for a fiber composite (in the form of a layer with two cylindrical solid inclusions) is proposed. However, the specified transition formulas in [27] do not take into account the relationship between the shifted cylindrical coordinate systems (relative to the Cartesian system of the layer), which does not allow solving problems with three or more stress concentrators. Additionally, works [21–24,26] study boundary conditions that do not satisfy the condition for modeling cylindrical supports.

From this point of view, the closest to the present work are publications [25,27], where displacements are set on the inner cylindrical surface of the pipe, or a rigid cylindrical inclusion is placed in the layer (under the condition of load balancing). This makes it possible to compare some of the results obtained in order to assess their reliability.

Thus, the problem of highly accurate determination of the stress state for a layer with N cylindrical indentations can be solved by employing the analytic-numerical generalized Fourier method. Infinite layer boundaries and longitudinal cavities will also be taken into account. However, in contrast to existing works, it is necessary to consider other types of boundary conditions and to apply formulas for the transition between the layer and the cylindrical coordinate systems shifted relative to the Cartesian system. Considering the necessity to calculate such models in practice, it is important and relevant to develop the most accurate methods for determining the stress–strain state for them.

The purpose of this paper is to:

1. develop a method for solving the problem for a layer with N cylindrical embedded supports based on the analytical-numerical Fourier method;

2. evaluate the influence of having more than one support on the stress–strain state of the body;

3. assess the impact of the distance between the supports on the stress–strain state of the body.

## 2. Materials and Methods

There is one layer located on N cylindrical embedded supports. Cylindrical embedded supports will be considered in local cylindrical coordinates $(\rho_p, \varphi_p, z)$, $p = 1\ldots N$. as cavities with zero displacements defined on their surface (Figure 1).

We will consider the layer in the Cartesian coordinate system $(x, y, z)$. We combine this Cartesian system with the coordinate system of the first cylinder $(p = 1, \overline{x}_1 = \overline{y}_1 = 0)$ and orient it equally. Distance to the center of the Cartesian coordinate system from the upper boundary of the layer $y = h$, from the lower boundary $y = -\widetilde{h}$. The surfaces of the

cylindrical cavities and the surface of the layer must be parallel to each other. The material of the layer is elastic, isotropic, and homogeneous, meaning it has linear elastic properties.

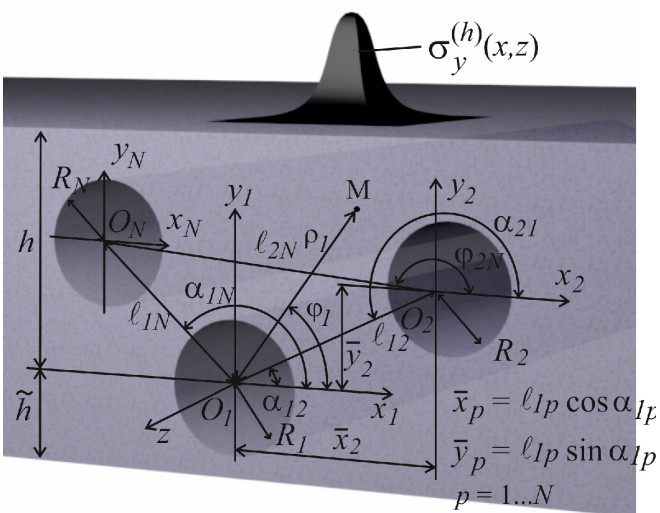

**Figure 1.** Layer with rigidly fixed cylindrical cavities.

The solution to the Lame equation needs to be found. The stresses $\overrightarrow{FU}(x,z)_{|y=h} = \overrightarrow{F}_h^0(x,z)$, $\overrightarrow{FU}(x,z)_{|y=-\tilde{h}} = \overrightarrow{F}_{\tilde{h}}^0(x,z)$ are specified on the boundaries of the layer, and the displacements $\overrightarrow{U}(\varphi_p,z)_{|\rho_p=R_p} = \overrightarrow{U}_p^0(\varphi_p,z)$ are specified on the boundaries of the cylindrical cavities, where

$$
\begin{aligned}
\overrightarrow{F}_h^0(x,z) &= \tau_{yx}^{(h)}\overrightarrow{e}_x + \sigma_y^{(h)}\overrightarrow{e}_y + \tau_{yz}^{(h)}\overrightarrow{e}_z \ , \\
\overrightarrow{F}_{\tilde{h}}^0(x,z) &= \tau_{yx}^{(\tilde{h})}\overrightarrow{e}_x + \sigma_y^{(\tilde{h})}\overrightarrow{e}_y + \tau_{yz}^{(\tilde{h})}\overrightarrow{e}_z \ , \\
\overrightarrow{U}_p^0(\varphi_p,z) &= U_\rho^{(p)}\overrightarrow{e}_\rho + U_\varphi^{(p)}\overrightarrow{e}_\varphi + U_z^{(p)}\overrightarrow{e}_z
\end{aligned}
\tag{1}
$$

$\tau_{yx}^{(h)}, \sigma_y^{(h)}, \tau_{yz}^{(h)}, \tau_{yx}^{(\tilde{h})}, \sigma_y^{(\tilde{h})}, \tau_{yz}^{(\tilde{h})}$ are the given functions of tangential and normal stresses on the corresponding boundary surface of the layer; $U_\rho^{(p)}, U_\varphi^{(p)}, U_z^{(p)}$ are the given displacement functions on the corresponding boundary surface of the cavity and along the corresponding coordinate axis; $\overrightarrow{e}_x, \overrightarrow{e}_y, \overrightarrow{e}_z$ are orts of the Cartesian coordinate system, $\overrightarrow{e}_\rho, \overrightarrow{e}_\varphi, \overrightarrow{e}_z$ are orts of the cylindrical coordinate system. For cylinders, these functions should decay rapidly to zero along the z axis. For layer boundaries, these functions should decay rapidly along the x and z coordinates.

We take the basic solutions of the Lamé equation in the form proposed in [15]. This allows us to obtain an exact solution for a single cylinder or layer. These basic solutions are as follows:

$$
\begin{aligned}
\overrightarrow{u}_k^{\pm}(x,y,z;\lambda,\mu) &= N_k^{(d)}e^{i(\lambda z+\mu x)\pm\gamma y}; \\
\overrightarrow{R}_{k,m}(\rho,\varphi,z;\lambda) &= N_k^{(p)}I_m(\lambda\rho)e^{i(\lambda z+m\varphi)}; \\
\overrightarrow{S}_{k,m}(\rho,\varphi,z;\lambda) &= N_k^{(p)}\left[(\text{sign}\lambda)^m K_m(|\lambda|\rho)\cdot e^{i(\lambda z+m\varphi)}\right]; k=1,\ 2,\ 3;
\end{aligned}
\tag{2}
$$

$$
\begin{aligned}
N_1^{(d)} &= \tfrac{1}{\lambda}\nabla; \quad N_2^{(d)} = \tfrac{4}{\lambda}(\nu-1)\overrightarrow{e}_2^{(1)} + \tfrac{1}{\lambda}\nabla(y\cdot); \\
N_3^{(d)} &= \tfrac{i}{\lambda}\text{rot}\left(\overrightarrow{e}_3^{(1)}\cdot\right); N_1^{(p)} = \tfrac{1}{\lambda}\nabla;
\end{aligned}
$$

$$N_2^{(p)} = \frac{1}{\lambda}\left[\nabla\left(\rho\frac{\partial}{\partial\rho}\right) + 4(\nu-1)\left(\nabla - \vec{e}_3^{(2)}\frac{\partial}{\partial z}\right)\right];$$

$$N_3^{(p)} = \frac{i}{\lambda}\mathrm{rot}\left(\vec{e}_3^{(2)}\cdot\right); \gamma = \sqrt{\lambda^2 + \mu^2}; -\infty < \lambda, \mu < \infty,$$

where $I_m(x)$, $K_m(x)$—modified Bessel functions; $\nu$—Poisson's ratio; $\vec{S}_{k,m}$, $\vec{R}_{k,m}$, k = 1, 2, 3—the external and internal solutions of the Lamé equation for cylindrical surfaces, respectively; $\vec{u}_k^{(+)}$—solution of the Lamé equation for the Cartesian coordinate system at y < 0; $\vec{u}_k^{(-)}$—solution of the Lamé equation for the Cartesian coordinate system at y > 0.

To switch between the basic solutions, we will use the following formulas proposed in [15,16], modified for the shifted cylindrical coordinate system ($p \neq 1$):

1. transform the basic solutions $\vec{S}_{k,m}$ of the cylindrical coordinate system shifted by $\overline{x}_p$ and $\overline{y}_p$ into the basic solutions $\vec{u}_k^{(-)}$ (at y > 0) and $\vec{u}_k^{(+)}$ (at y < 0) of the Cartesian coordinate system

$$\vec{S}_{k,m}\left(\rho_p, \varphi_p, z; \lambda\right) = \frac{(-i)^m}{2}\int_{-\infty}^{\infty}\omega_{\mp}^m \cdot e^{-i\mu\overline{x}_p \pm \gamma\overline{y}_p} \cdot \vec{u}_k^{(\mp)} \cdot \frac{d\mu}{\gamma}, k = 1, 3;$$

$$\vec{S}_{2,m}\left(\rho_p, \varphi_p, z; \lambda\right) = \frac{(-i)^m}{2}\int_{-\infty}^{\infty}\omega_{\mp}^m \cdot \left(\left(\pm m\cdot\mu - \frac{\lambda^2}{\gamma} \pm \lambda^2\overline{y}_p\right)\vec{u}_1^{(\mp)} \mp \right.$$

$$\left. \mp\lambda^2\vec{u}_2^{(\mp)} \pm 4\mu(1-\nu)\vec{u}_3^{(\mp)}\right) \cdot \frac{e^{-i\mu\overline{x}_p \pm \gamma\overline{y}_p}d\mu}{\gamma^2}, \tag{3}$$

where $\omega_{\mp}(\lambda, \mu) = \frac{\mu\mp\gamma}{\lambda}$; $m = 0, \pm 1, \pm 2, \ldots$; $\overline{x}_p, \overline{y}_p$– coordinates of the shifted local cylindrical coordinate system (Figure 1);

2. transition from basic solutions $\vec{u}_k^{(+)}$ and $\vec{u}_k^{(-)}$ of the Cartesian coordinate system to basic solutions $\vec{R}_{k,m}$ of the cylindrical coordinate system

$$\vec{u}_k^{(\pm)}(x, y, z) = e^{i\mu\overline{x}_p \pm \gamma\overline{y}_p} \cdot \sum_{m=-\infty}^{\infty}(i\cdot\omega_{\mp})^m\vec{R}_{k,m}, (k = 1, 3);$$

$$\vec{u}_2^{(\pm)}(x, y, z) = e^{i\mu\overline{x}_p \pm \gamma\overline{y}_p} \cdot \sum_{m=-\infty}^{\infty}\left[(i\cdot\omega_{\mp})^m \cdot \lambda^{-2}\left(\left(m\cdot\mu + \overline{y}_p\cdot\lambda^2\right)\cdot\vec{R}_{1,m}+\right.\right.$$

$$\left.\left. +4\mu(1-\nu)\vec{R}_{3,m} \pm \gamma\cdot\vec{R}_{2,m}\right)\right], \tag{4}$$

where $\vec{R}_{k,m} = \widetilde{\vec{b}}_{k,m}\left(\rho_p, \lambda\right)\cdot e^{i(m\varphi_p + \lambda z)}$; $\widetilde{\vec{b}}_{1,n}(\rho, \lambda) = i\cdot I_n(\lambda\rho)\cdot\left(\vec{e}_\varphi\frac{n}{\lambda\rho} + \vec{e}_z\right) + \vec{e}_\rho \cdot I'_n(\lambda\rho)$;

$\widetilde{\vec{b}}_{2,n}(\rho, \lambda) = \vec{e}_\varphi i\cdot m\left(I'_n(\lambda\rho) + \frac{4(\nu-1)}{\lambda\rho}I_n(\lambda\rho)\right) + \vec{e}_\rho\cdot[(4\nu-3)\cdot I'_n(\lambda\rho) + \lambda\rho I''_n(\lambda\rho)] + $

$+\vec{e}_z i\lambda\rho I'_n(\lambda\rho)$;

$\widetilde{\vec{b}}_{3,n}(\rho, \lambda) = -\left[\vec{e}_\varphi\cdot i\cdot I'_n(\lambda\rho) + \vec{e}_\rho\cdot I_n(\lambda\rho)\frac{n}{\lambda\rho}\right]$;

3. for transition from the basic solutions of the cylinder with index $p$ to the basic solutions of the cylinder with index $q$

$$\vec{S}_{k,m}\left(\rho_p, \varphi_p, z; \lambda\right) = \sum_{n=-\infty}^{\infty}\vec{b}_{k,pq}^{mn}\left(\rho_q\right)\cdot e^{i(n\varphi_q + \lambda z)}, k = 1, 2, 3 \tag{5}$$

$$\vec{b}_{1,pq}^{mn}\left(\rho_q\right) = (-1)^n\widetilde{\vec{b}}_{1,n}\left(\rho_q, \lambda\right)\widetilde{K}_{m-n}\left(\lambda\updownarrow_{pq}\right)\cdot e^{i(m-n)\alpha_{pq}};$$

$$\vec{b}_{3,pq}^{mn}\left(\rho_q\right) = (-1)^n \widetilde{K}_{m-n}\left(\lambda\updownarrow_{pq}\right)\cdot e^{i(m-n)\alpha_{pq}}\cdot\vec{b}_{3,n}\left(\rho_q,\lambda\right);$$

$$\vec{b}_{2,pq}^{mn}\left(\rho_q\right) = (-1)^n\left\{\vec{b}_{2,n}\left(\rho_q,\lambda\right) - \tfrac{\lambda}{2}\updownarrow_{pq}\cdot\widetilde{K}_{m-n}\left(\lambda\updownarrow_{pq}\right)\cdot\right.$$

$$\left.\cdot\left[\widetilde{K}_{m-n+1}\left(\lambda\updownarrow_{pq}\right) + \widetilde{K}_{m-n-1}\left(\lambda\updownarrow_{pq}\right)\right]\cdot\vec{b}_{1,n}\left(\rho_q,\lambda\right)\right\}\cdot e^{i(m-n)\alpha_{pq}},$$

where $\alpha_{pq}$—the angle from the $x_p$ axis to the segment $\updownarrow_{qp}$; $\widetilde{K}_m(x) = (sign(x))^m\cdot K_m(|x|)$; $q = 1\dots N$; $p \neq q$.

The parallel shifted cavity $q$ is located relative to the parallel shifted cavity $p$ at a distance and angle:

$$L_{pq} = \left|\begin{array}{l}\sqrt{L_{1p}^2 + L_{1q}^2 - 2\cdot L_{1p}\cdot L_{1q}\cdot\cos(\alpha_{1q} - \alpha_{1p})},\text{ at }\alpha_{1q}\geq\alpha_{1p}\\[2mm]\sqrt{L_{1p}^2 + L_{1q}^2 - 2\cdot L_{1p}\cdot L_{1q}\cdot\cos(\alpha_{1p} - \alpha_{1q})},\text{ at }\alpha_{1q}<\alpha_{1p}\end{array}\right., \qquad (6)$$

$$\alpha_{pq} = \left|\begin{array}{l}\alpha_{1p} - \arccos\left(\frac{L_{1p}^2 + L_{pq}^2 - L_{1q}^2}{2\cdot L_{1p}\cdot L_{pq}}\right) + \pi,\text{ at }\alpha_{1q}\geq\alpha_{1p}\\[3mm]\alpha_{1p} - \arccos\left(\frac{L_{1p}^2 + L_{pq}^2 - L_{1q}^2}{2\cdot L_{1p}\cdot L_{pq}}\right) - \pi,\text{ at }\alpha_{1q}<\alpha_{1p}\end{array}\right.$$

### 3. Results

*3.1. Create and Solve a System of Equations to Determine the Unknowns of the Lamé Equation*

The solution to the problem is represented in the form:

$$\vec{U} = \sum_{k=1}^{3}\int_{-\infty}^{\infty}\int_{-\infty}^{\infty}\left(H_k(\lambda,\mu)\cdot\vec{u}_k^{(+)}(x,y,z;\lambda,\mu) + \widetilde{H}_k(\lambda,\mu)\cdot\vec{u}_k^{(-)}(x,y,z;\lambda,\mu)\right)d\mu\,d\lambda +$$
$$+ \sum_{p=1}^{N}\sum_{k=1}^{3}\int_{-\infty}^{\infty}\sum_{m=-\infty}^{\infty}B_{k,m}^{(p)}(\lambda)\cdot\vec{S}_{k,m}\left(\rho_p,\varphi_p,z;\lambda\right)d\lambda, \qquad (7)$$

where $\vec{R}_{k,m}\left(\rho_p,\varphi_p,z;\lambda\right)$, $\vec{S}_{k,m}\left(\rho_p,\varphi_p,z;\lambda\right)$, $\vec{u}_k^{(-)}(x,y,z;\lambda,\mu)$ and $\vec{u}_k^{(+)}(x,y,z;\lambda,\mu)$ basic solutions (2); $H_k(\lambda,\mu)$, $\widetilde{H}_k(\lambda,\mu)$ i $B_{k,m}^{(p)}(\lambda)$—unknown functions to be found from the boundary conditions (1); $p$—cylindrical cavity number.

To assemble the system of equations, we satisfy the boundary conditions at the upper and lower boundaries of the layer. To achieve this, the vectors $\vec{S}_{k,m}$ in the right side of (7) are rewritten in the layer's coordinate system by the basic solutions $\vec{u}_k^{(-)}$ at $y = h$ and $\vec{u}_k^{(+)}$ at $y = -\widetilde{h}$. This is possible if we use the transition Formula (3). For the resulting vectors, we find the stresses and equate them at $y = h$ to the given $\vec{F}_h^0(x,z)$, at $y = -\widetilde{h}$ to the given $\vec{F}_{\widetilde{h}}^0(x,z)$, after representing them using a double Fourier integral.

From this system of equations described above, we find the functions $\widetilde{H}_k(\lambda,\mu)$ and $H_k(\lambda,\mu)$ by $B_{k,m}^{(p)}(\lambda)$.

Now, we will collect the system of equations for cylindrical cavities. To perform this, we rewrite the right-hand side of (7) in the local cylindrical coordinate system for each cavity $p$ separately. This is possible if, using the transition Formula (4), we rewrite $\vec{u}_k^{(+)}(x,y,z;\lambda,\mu)$ and $\vec{u}_k^{(-)}(x,y,z;\lambda,\mu)$ in terms of the basis solutions of $\vec{R}_{k,m}$, and, using the transition Formula (5), for each cylinder $p \neq q$, we rewrite $\vec{S}_{k,m}\left(\rho_p,\varphi_p,z;\lambda\right)$ in terms of the basis solutions of $\vec{R}_{k,m}$. The resulting vector, at $\rho p = R_p$, is equal to zero (zero displacements in the supports are set). From the resulting system of equations, we replace

the previously found functions $\widetilde{H}_k(\lambda, \mu)$ and $H_k(\lambda, \mu)$ with these functions found using $B_{k,m}^{(p)}(\lambda)$.

After eliminating the series over $m$ and integrals over $\lambda$, we obtain a set of $3N$ infinite systems of linear algebraic equations to determine the unknowns $B_{k,m}^{(p)}(\lambda)$.

The following reduction method can be applied to the obtained systems, which exhibit fast convergence of approximate solutions to the exact ones, as evidenced by the results of fulfilling the boundary conditions in a numerical study. In addition, similar systems and the implementation of boundary conditions have been studied in detail in [16,26].

After finding the functions $B_{k,m}^{(p)}(\lambda)$, substitute them into the expressions for $H_k(\lambda, \mu)$ and $\widetilde{H}_k(\lambda, \mu)$. This will determine all the unknowns.

### 3.2. Numerical Analysis of the Stress State of the Layer

The problem is solved for a layer and three cylindrical embedded supports represented as cavities (Figure 1). The cylindrical coordinate system of the cavity $p = 1$ coincides with the Cartesian coordinate system of the layer. The cavity $p = 2$ is located at a distance $L_{12} = 30$ mm, at an angle $\alpha_{12} = 0$. The cavity $p = 3$ is located at $L_{13} = 30$ mm, $\alpha_{13} = \pi$.

Layer: ABS plastic, modulus of elasticity E = 1700 MPa, Poisson's ratio $\nu$ = 0.38. The radius of the cylindrical cavities (embedded supports) $R_1 = R_2 = R_3 = 5$ mm. Distance from the line of the supports to the upper and lower boundaries of the layer $h = \widetilde{h} = 12$ mm.

At the upper boundary of the layer, stresses are set in the form of a wave (Figure 1) $\sigma_y^{(h)}(x,z) = -10^8 \cdot (z^2 + 10^2)^{-2} \cdot (x^2 + 10^2)^{-2}$, $\tau_{yx}^{(h)}(x,z) = \tau_{yz}^{(h)}(x,z) = 0$. Zero stresses are set at the lower boundary of the layer $\sigma_y^{(\widetilde{h})}(x,z) = \tau_{yx}^{(\widetilde{h})}(x,z) = \tau_{yz}^{(\widetilde{h})}(x,z)$.

For comparison, a variant with one cavity (support), with three cavities (supports) with a distance between them of $L_{12} = L_{13} = 15$ mm, and with the load shifted to the right between the supports, was calculated.

To obtain numerical values, the infinite system of equations was replaced by a finite truncated system up to $m = 5$. With the given geometric parameters and load value for values from zero to one, it was possible to satisfy the boundary conditions with an accuracy of $10^{-4}$.

In Figure 2, the stress distribution $\sigma_\rho, \sigma_\varphi, \sigma_z, \tau_{\rho\varphi}$ on the surface of the cavity p = 1 at $z = 0$, $L_{12} = L_{13} = 30$ mm.

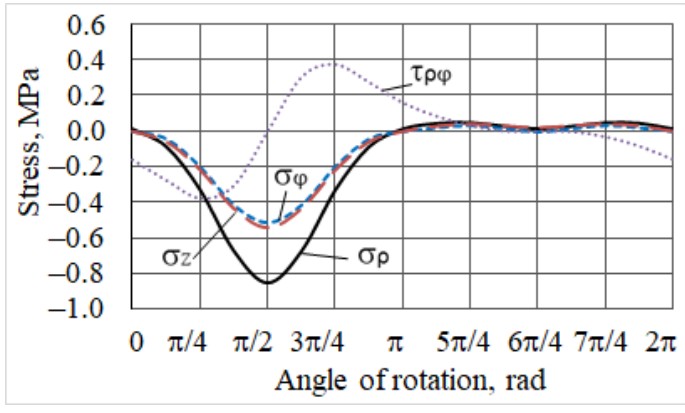

**Figure 2.** Stress distribution on the surface of the cavity $p = 1$.

Figure 2 shows that the stresses are concentrated in the upper part of the cavity surface. This is due to the location of the load above this support and its rigid fixation. For a maximum load value of $\sigma_y = 1$ MPa, the maximum stress values are $\sigma_\rho = -0.8528$ MPa, $\sigma_\varphi = -0.5159$ MPa, $\sigma_z = -0.5412$ MPa, $\tau_{\rho\varphi} = 0.3758$ MPa.

In Figure 3, the distribution of stress $\sigma_\varphi$ on the surface of cavity $p = 1$ at $z = 0$ is shown for different distances between the inclusions.

When the distance between the supports is reduced, the stress $\sigma_\varphi$ on the surface of the cavity $p = 1$ decreases (Figure 3). Thus, the maximum stress at $L_{12} = L_{13} = 15$ mm is equal to $\sigma_\varphi = -0.48516$ MPa, which is 6% lower than $\sigma_\varphi = -0.51593$ MPa (at $L_{12} = L_{13} = 30$ mm). The decrease in stress on the cavity $p = 1$ is due to its redistribution on the surface of the approaching cavities.

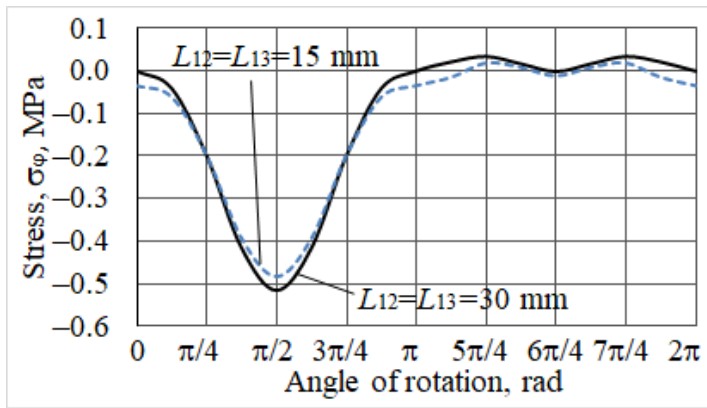

**Figure 3.** Stress $\sigma_\varphi$ on the surface of the cavity $p = 1$.

Figure 4 shows the distribution of tangential stresses $\tau_{\rho\varphi}$ on the surface of the cavity $p = 1$ at $z = 0$ and different distances between the supports.

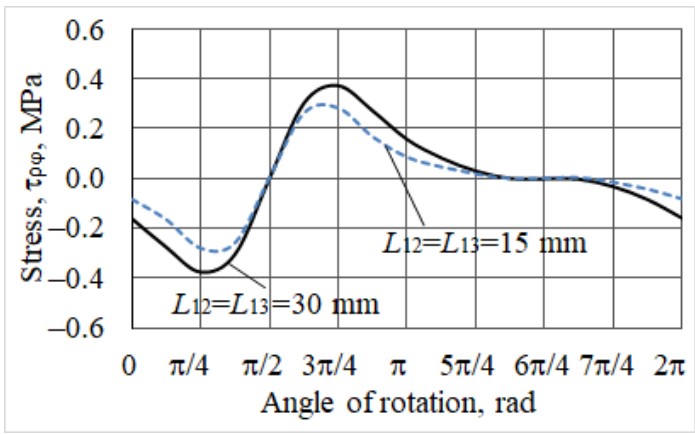

**Figure 4.** Stress $\tau_{\rho\varphi}$ on the surface of the cavity $p = 1$.

Figure 4 shows that the distance between the supports has a slightly greater effect on the tangential stresses. Thus, at $L_{12} = L_{13} = 15$ mm, the maximum stresses are equal to $\tau_{\rho\varphi} = +0.2828$ MPa, and the difference is 7.5% (at $L_{12} = L_{13} = 30$ mm $\tau_{\rho\varphi} = +0.3758$ MPa).

When calculating a layer with one support [27], the maximum stress is $\sigma_\varphi = -0.521$ MPa, and the tangential stress is $\tau_{\rho\varphi} = +0.381$ MPa. This is an increase of 1% for $\sigma_\varphi$ and 1.4% for $\tau_{\rho\varphi}$ compared to the case with $L_{12} = L_{13} = 30$ mm. This means when the distance between the supports is greater than $L_{12} = L_{13} = 30$ mm, the effect on the stress state of the support $p = 1$ is insignificant, and the calculation with such a geometric arrangement of the load can be carried out with only one support.

If, at $L_{12} = L_{13} = 30$ mm, the load is shifted to the right between supports $p = 1$ and $p = 2$, the stresses $\sigma_\varphi$ on these cavities will have the form shown in Figure 5.

When the load is placed between the supports, the maximum stresses $\sigma_\varphi$ on these supports (Figure 5) are equal to $\sigma_\varphi^1 = -0.265$ MPa and $\sigma_\varphi^2 = -0.2621$ MPa (1% difference). This indicates that when the load is located between two supports, the influence of the third support on the stress state is insignificant.

Figure 6 shows the stress $\sigma_x$ distribution at the upper boundary of the layer at $L_{12} = L_{13} = 30$ mm, $L_{12} = L_{13} = 15$ mm, and the variant where the load is located between the supports.

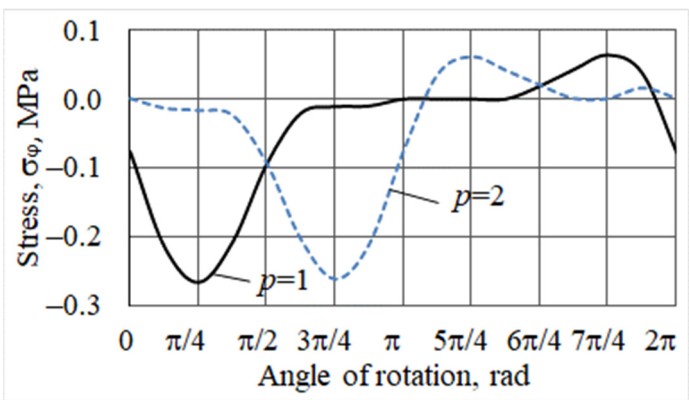

**Figure 5.** Stresses $\sigma_\varphi$ on the cavity surfaces $p = 1$ and $p = 2$.

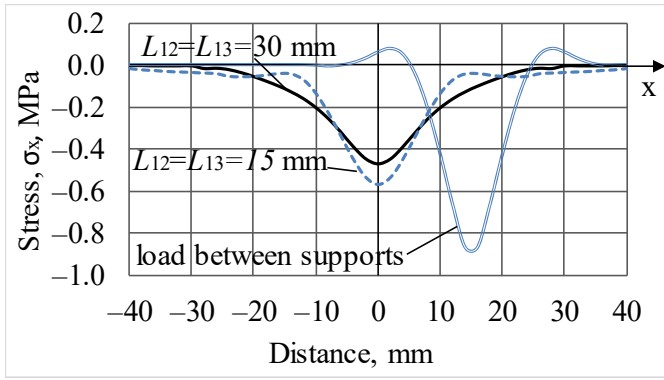

**Figure 6.** The stress $\sigma_x$ at the top of the layer.

Figure 6 shows that when the supports are approached, the maximum stress $\sigma_x$ at the top of the layer increases and is maximum in the zone of load application $\sigma_x = -0.567968$ MPa. Shifting the load to the zone between supports $p = 1$ and $p = 2$, with $L_{12} = L_{13} = 30$ mm, results in an increase of the maximum stress in the load application zone $\sigma_x = -0.94884$ MPa (with the specified maximum $\sigma_y = 1$).

## 4. Discussion

A high-precision method for solving the problem for a layer with N circular cylindrical embedded supports has been proposed. In difference from the existing works [17–27], which also apply the generalized Fourier method, we used other types of boundary conditions (1), and the transition formulas modified for the shifted cylindrical coordinate systems (3)–(6) are taken into account.

The stress state for a layer with three embedded supports at different distances between the supports and different load locations is numerically investigated.

The analysis of the stress state allows us to assert:

1. when the load is located above the support, the influence of other supports on the stress state becomes noticeable at the distance between the supports $\frac{R_1 + R_2}{L_{12}} > 0.33$;

2. when the supports are close to each other, the stress on the surface of the cavity decreases, but the stress at the layer boundary increases;

3. when the load is located between the supports, the stresses on the supports decrease but increase significantly at the layer boundary.

The proposed method can be used for high-precision calculations of the stress state of structures, machine parts, and mechanisms whose model is a layer on cylindrical embedded

supports or a layer with parallel cylindrical cavities with specified displacements in these cavities.

Further development of this research is necessary for hinged supports, which are also frequently encountered in practical calculations.

**Author Contributions:** Conceptualization, V.M.; methodology, V.M.; software, V.S.; validation, V.N., O.S. and V.S.; formal analysis, V.N.; investigation, V.M.; resources, O.S.; data curation, V.M.; writing—original draft preparation, V.M.; writing—review and editing, V.M. and V.N.; visualization, O.S.; supervision, V.M.; project administration, V.M. All authors have read and agreed to the published version of the manuscript.

**Funding:** This research received no external funding.

**Data Availability Statement:** Not applicable.

**Conflicts of Interest:** The authors declare no conflict of interest.

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
