# Peer review of "Solving the Problem of Elasticity for a Layer with N Cylindrical Embedded Supports"

_computation, doi:10.3390/computation11090172_

Round 1

Reviewer 1 Report

Dear Authors,

Thank you for submitting your paper to Computation. Your article presents your solution for a layer located on cylindrical supports. Analytical solution for any problem is precious if it is verified. In the introduction, you criticized approximate methods including the finite element method. But I think it is easy to obtain precise solutions for problems presented in section 3 using the finite element method. So can you stress out the advantages and drawbacks of your approach regarding the finite element method?

Figure 3 is basic for your description. But it is not good. It presents very poorly the model in the thickness direction and should be improved. Symbols used in the drawing are not consistent with symbols in your formulas.

Fig. 3 is also not suitable for your case study. As I understand the description of your problem the supports are located in a line so Fig. 3  in this case is misleading. It is not very strenuous to attach the new drawing for better visualization of the problem.

In the text, there are problems with Ukrainian letters. They appear in eq. 6 and in line 185.

You refer to 29 but there is no such reference in the list of references.

Best regards,

Reviewer 2 Report

Article present very interesting topic and contains new items. Author presents a method for solving the problem for a layer with N cylindrical embedded supports based on the analytical-numerical Fourier method. What is more the proposed method for calculating the stress-strain state can be used in the design of structures whose design scheme is a layer with cylindrical embedded supports. The used research methods seems to be appropriate and analysis was performed properly.

I can recommend this article for publish in current form.

Author Response

Thank you for studying my article.

Reviewer 3 Report

The reviewed here manuscript is entitled “Solving the Problem of Elasticity Theory For a Layer With N Cylindrical Embedded Supports” and presents an analysis based on analytical- numerical solution of the stress distribution in an elastic layer with embedded rigid cylindrical supports. First comment is to the title of the manuscript – it contains unnecessary information which makes it difficult to perceive. My suggestion is to reduce the title to “Solving the Problem of Elasticity for a Layer with Cylindrical Embedded Supports”.

The manuscript is written in very poor English, not so much in grammar as in scientific terminology and inappropriate use of words that have a similar meaning. In addition, there are unfinished expressions, all of which make reading the manuscript very difficult. My strong suggestion is that the manuscript be edited primarily for the English language by a specialist in the field or a professional translator and then resubmitted for peer review. A list of symbols and a reconciliation of the symbols used in the graphical representation of the problem under consideration and those in the text and formulas are also required. Some of the problematic wording, but not all, as there are too many, are given below.

In Abstract:

L10: The main goal of “deformable solid mechanics”… - “solid mechanics” is the English term.

L12: “However, not all design schemes can be solved using exact methods” – unclear meaning of “design scheme” in this statement, e.g. loading diagram or just structute/construction.

L16: “The basic solutions of the Lamé equation between these coordinate systems are related using the generalized Fourier method.” – messy statement, needs attention.

L19: “The numerical analysis has shown high accuracy of the boundary conditions” -- ..Do you mean “ high accuracy in satisfying the boundary conditions “

L20: “and the physical regularity of the stress distribution and the comparison with other tested works indicates the reliability of the results obtained” – what is the meaning of “physical regularity” and “other tested works”?

In Introduction (only few of the problematic phrases, Introduction must be carefully rewritten to be readable and understandable).

L31: “most spatial design schemes (primarily those with more than three boundary surfaces) do not allow for accurate methods.” – messy statement; “spatial” is usually replaced by 3D; what exactly is the meaning of “accurate method” here, e.g. analytical or exact solution. In engineering, inaccurate methods should not be used to solve problems and the accuracy is always predefined if approximate solutions are the only ones possible..  

L35: “Therefore, to improve accuracy, tests of finished elements are performed [2] or a combination of different calculation methods [3].” – it seems to be a messy automatic translation – how performing tests can improve the accuracy of a solution? -  and the phrase is not completed e.g “… a combination of … methods is used in [3]”. Missing verbs is an often met problem in the manuscript.

L44: “cylindrical absolutely rigid inclusions” – … perfectly rigid… or just “rigid”

L57: “a layered composite is considered for impact loading”… - a layered composite under impact loading is considered… “for impact loading” is unclear.

L60: “In [9], a simple analytical problem in the form of a trigonometric series is formed for a similar model.” – unclear statement, needs attention (moreover, a problem is formulated, posed etc. but not formed)

L71: “solution in plane polar coordinates” what is the reason to specify the polar coordinates are plane coordinates?

L81: “The most effective method for this problem” – specify the problem at this place. This kind of problematic formulation is common. The text has to be checked for such unclear references.

In 2. Materials and Methods of studying the stress state of the layer – long title and messy in meaning – materials are clear from the title – elastic, may be linear elastic, homogeneous, isotropic… Title as “Materials and Methods” …is well explanatory and not misleading.  

The layer is not located but resting on some cylinders.

L139: “…which is equally oriented and combined with the coordinate system of the first cylinder .. “ to specify which one is the “first cylinder”. How to equally orient angles with lines? What is combined in this case?

L150: “The basic solutions of the Lamé equation are in the form [15].” – unclear what is the “form” – e.g. the form is given in [5] or “have the following form” and give reference to the corresponding equations.

L153: “gamma” is an example of repeatedly explained symbol. The text has to be checked and make free of repeated explanations.

L158: “To switch between the basic solutions, we will use the formulas [15, 16], modifying them for the shifted cylindrical coordinate system:” .. do you mean “… we will use the following formulas…” ? The sense of “shifted coordinates” needs clarification.

L179: Russian words to be removed in (6).

In 3. Results the study of the stress state – the title is unclear. Simply “Results” is well explanatory.

3.1. Creating and solving a system of equations – imprecise title; it is not any system that is behind “ a system of equations”. Reconsider the title of this subsection.

L188: “To form the system of equations, we satisfy the boundary conditions at the upper and lower boundaries of the layer.” – the system is completed or assembled but not “formed”, to complete the system you have “”imposed the boundary conditions (this will complete your boundary and initial problem). The usage of “form” within the text needs attention.

l194: “From this system of equations…” specify what system

l199- 201 – long and messy sentence, to be reconsidered.

L202: “From the resulting system of equations, we exclude the previously found functions” – In what sense the functions are “excluded”?

L212: “…which exhibits fast convergence of approximate solutions to the exact ones.” You can claim the fast convergence of the solution; it is to prove whether it converges to the exact solution or not.

L218 “stresses are set in the form of a wave” – there is no sense in this statement, better to be deleted as the applied stress is explicitly given with a formula.

L224: “The infinite system of equations was reduced by the parameter m = 5,” – unclear statement; in practice this system is replaced by a finite truncated system, in this case up to m=5.

L234: “This is due to the redistribution of stress to the approaching cavities.” – unclear, needs attention

Figure 4. The stresses  on the surface of the cavity are p = 1 and p = 2. --- the figure caption is unclear, needs attention.

In 4. Discussion

The statement l. 270 -272 is unclear, needs to be reformulated.

L278: -…  when the supports approach, the stress on the surface of the cavity decreases, but the stress at the layer boundary increases;” – requires attention to be reformulated into correct English as in the present form is not readable.

L 285 Further development of this area is necessary  -  what area?; consider to use subject instead of area, or rewrite the sentence appropriately.

My strong suggestion is that the manuscript be edited primarily for the English language by a specialist in the field or a professional translator and then resubmitted for peer review. I listed some of the problematic issues in my review report.

Reviewer 4 Report

The manuscript is currently far from being ready for publication and needs to be revised throughout. In the revision, except for the following comments, the question and its research significance need to be clearly stated, all equations are either derived or given their sources, and all variables must be explained as soon as they first appear.

1. There is a clerical error, “3”, in line 2.

2. I am not sure what the “such calculations” in line 11 stands for.

3. I am not sure what the “such schemes” in lines 12-13 means.

4. It is suggested to change the “-” in lines 125, 127, 129, 160, 164, 171, 276, 278 and 280 to a number.

5. The “Materials and Methods of studying the stress state of the layer” in line 131, the title of Section 2 needs to be rewritten.

6. The number of cylindrical embedded supports in Figure 1 is 3, not the “N” in line 132.

7. Figure 1 is unclear, especially the relation between local cylindrical coordinates and Cartesian coordinate system, and needs to be redrawn. The description of Figure 1 is so poor that it needs to be rewritten.

8. In Figure 1, which one does the “the first cylinder” in line 140 mean?

9. I am not sure what the “The surfaces of the cylindrical cavities and the surface of the layer must not intersect.” in lines 141-142 means.

10. Please give the basis or source of Equation (1).

11. Please rewrite the “are known functions” in line 148.

12. Please rewrite the “in the form [15]” in line 150.

13. The “there”in lines 155, 162, 176 and 185, should be “where”.

14. Please rewrite the “the formulas [15, 16]” in line 158.

15. There is no the “coordinates of the shifted local cylindrical coordinate system (Fig. 1)” in lines 162-163 in Figure1.

16. Please check Equation (6) in line 180 carefully.

17. The “Results the study of the stress state” in line 182, the title of Section 3 needs to be rewritten.

18. The “3.1” in line 210 should be “3.2”.

19. The “29” in line 269 should be “28”.

20. Please rewrite the “whose design scheme is a layer on cylindrical embedded supports or a layer with cylindrical cavities and displacements specified on the cavities” in lines 283-285.

21. Once again, all variables must be explained as soon as they first appear.

I hope my next Recommendation is not “Reject”.

Minor editing of English language required.

Round 2

Reviewer 1 Report

Dear Authors,

Thank you for addressing all the doubts. I discovered I had accidentally mistaken the figure number but you understand my point correctly.

Best regards,

Author Response

Thank you for studying my article.

Reviewer 3 Report

Minor corrections still needed:

"and the physical regularity of the stress distribution and the compari-20 son with other tested works indicates the reliability of the results obtained" - In your explanation to my remark you stated that "other tested works" are "works close to the presented ", therefore, you should write , e.g.,  "... comparison with results of similar studies" instead of "comparison with other tested works", because "tested works" has different meaning, it means works that are verified.

My comment about using English is that professional help is needed to achieve good readability, as the reader will not see your explanations to the reviewers of what the problematic phrases mean. The literal translation from Russian to English, as used in many places in the text, can only be intuitively understood by Russian speakers.

Formatting of references should be in accordance with the journal's requirements, e.g. capital letters in the publication title, etc.  (example: "Investigation of the stress state of a composite in the form of a layer and a half space with a longitudinal 554 cylindrical cavity at stresses given on boundary surfaces." has to be "Investigation of the Stress State of a Composite ...")

My comment about using English is that professional help is needed to achieve good readability, as the reader will not see your explanations to the reviewers of what the problematic phrases mean. The literal translation from Russian to English, as used in many places in the text, can only be intuitively understood by Russian speakers.

Example: "... comparison with results of similar studies"should replace "comparison with other tested works" if the Authors want to state that a comparison is done with results from literature, because "tested works" has different meaning

Author Response

The corrections have been taken into account. Thank you for your review and help in improving the article.

Reviewer 4 Report

The manuscript has been sufficiently improved and now can be accepted for publication in Computation.

Author Response

Thank you for studying my article.